# Long GHz-Burst Laser Surface Polishing of AlSl 316L Stainless Steel Parts Manufactured by Short GHz-Burst Laser Ablation

**DOI:** 10.3390/nano15171343

**Published:** 2025-09-01

**Authors:** Théo Guilberteau, Florent Husson, Manon Lafargue, John Lopez, Marc Faucon, Laura Gemini, Inka Manek-Hönninger

**Affiliations:** 1Université de Bordeaux-CNRS-CEA, CELIA UMR 5107, 33405 Talence, France; 2ALPhANOV, Rue François Mitterrand, 33400 Talence, France; 3Amplitude, Cité de la Photonique, 11 Avenue de Canteranne, 33600 Pessac, France

**Keywords:** GHz-burst mode, laser processing, metals, stainless steel, polishing, ablation

## Abstract

GHz-burst laser polishing is as a promising technique for improving the surface quality of metallic materials, offering key advantages over conventional methods. In this study, two distinct approaches are investigated: a single-step polishing process, and a double-step process consisting of an initial laser milling step followed by a finishing/polishing pass. This distinction is critical in evaluating the performance of GHz-burst regimes under different surface conditions and roughness levels. Initial proof-of-concept trials confirm that GHz-burst irradiation can significantly reduce the surface roughness with minimal thermal damage, provided that process parameters are carefully optimized. Further analysis of spot-to-spot overlap reveals that the deposited energy density plays a crucial role in achieving uniform surface quality without inducing surface defects. The number of passes is also studied, showing that while multiple passes can improve surface finish, the benefit strongly depends on the initial roughness state of the substrate. Scalability is demonstrated by increasing both the repetition rate and scan speed proportionally while maintaining processing quality across larger areas. These results support the viability of GHz-burst laser polishing for high-throughput manufacturing. Applications in aerospace, biomedical implants, and precision optics highlight the technique’s potential for industrial adoption in demanding surface finishing contexts.

## 1. Introduction

Surface finishing is a critical step in the manufacturing of high-precision components in various industries [1], including aerospace [2], medical devices [3], electronics [4], and optical systems [5]. Surface quality significantly impacts mechanical performance, fatigue life, corrosion resistance, and optical properties [6,7,8]. Several polishing mechanisms are commonly employed to achieve the desired surface roughness, each with its advantages and limitations.

Mechanical polishing, for instance, involves abrasive processes in which material is removed through friction and shear, often leading to sub-surface damage, residual stresses, and non-uniformity of complex geometries [9]. Chemical-mechanical polishing (CMP), which is widely used in semiconductor manufacturing, offers better uniformity and precision but requires extensive process control and can introduce chemical contamination [10,11,12]. Electrochemical polishing provides smooth oxide-free surfaces, but is typically restricted to conductive materials and involves hazardous electrolytes [13]. Plasma and ion beam polishing are effective for achieving ultra-smooth surfaces but are costly and time-intensive, limiting their scalability for industrial applications [14,15,16].

On the other hand, laser-based polishing presents a contactless, flexible, and highly controllable alternative [17,18]. The principle of laser polishing is to heat the material surface locally until it reaches its melting point. At this temperature, the surface layer becomes a thin liquid film; due to surface tension forces, the molten metal flows and redistributes to fill valleys and smooth out peaks, effectively reducing the surface roughness. By precisely tuning the laser parameters, this approach can achieve high surface quality without equipment wear or chemical use [19,20].

An additional significant advantage of laser polishing is that it is a single-step finishing process that can often be performed on the same machine tool used for previous manufacturing steps, such as milling in injection molding tool production or direct energy deposition (DED) in additive manufacturing of metallic parts. This integration eliminates the need to reposition the part between different machines, avoiding the time-consuming and error-prone step of matching the reference of the machine with the coordinate system of the part, thereby increasing the overall process efficiency and precision.

However, conventional laser polishing methods are often limited by low throughput and thermal effects [21,22], which can lead to surface defects such as micro-cracking, oxidation, and heat-induced roughness.

In recent years, the field of laser polishing has seen notable advancements with the introduction of GHz-burst laser technology. Prior to this, the development of industrial high-power femtosecond (fs) laser sources marked a significant leap in precision and efficiency. For metals, which exhibit linear absorption, femtosecond lasers still provide significant benefits thanks to their ultrashort pulse durations. These pulses deliver energy so rapidly that heat diffusion into the surrounding material is minimized, reducing thermal damage and residual mechanical stress compared to longer pulse durations [23]. This precise energy control improves the polishing quality on metallic surfaces.

MHz-burst [24] and GHz-burst [25,26,27] laser technology builds on these advances. Differing from traditional continuous or pulsed laser systems, these techniques deliver energy in tightly packed pulse trains called bursts. The short pulse-to-pulse delay within the bursts enables controlled heat accumulation between pulses, producing a beneficial cooperative effect that enhances ablation efficiency [27,28,29,30,31]. While this process remains fundamentally thermal, precise control of energy delivery allows for improved surface quality, especially for metals such as 316L stainless steel, where high precision is essential. A recent publication has addressed this topic using short bursts [32]. Rather than emphasizing minimized heat-affected zones, it is more accurate to highlight the controlled thermal interaction that enables efficient polishing with refined surface finishes.

In this work, we show a proof-of-concept and investigate the scalability of long GHz-burst laser polishing for 316L stainless steel, focusing on the relationship between processing parameters such as the pulse-to-pulse and line-to-line overlap, number of passes, burst repetition rate, number of pulses in the burst and cumulative dose, and final surface roughness. Our study provides critical insight into the practical limits and potential optimization of high-throughput high-quality surface finishing.

## 2. Materials and Methods

### 2.1. Laser System

The laser system used for this experiment was a Tangor 100 from Amplitude (Amplitude Laser Group, Pessac, France), which emits pulses at 1030 nm wavelength with a pulse duration of 505 fs. The output beam is linearly polarized. This laser system can operate in repetitive single-pulse (SP) mode, in MHz-burst mode with up to 32 pulses per burst (ppb) at a 40 MHz repetition rate, or in GHz-burst mode with up to 800 ppb at an intra-burst repetition rate of 1.28 GHz. The burst repetition rate is adjustable up to 800 kHz and the pulse/burst energy available on the target ranges from 0 to 300 µJ.

For this study, we selected bursts of 50 and 800 pulses per GHz-burst with uniform intensity. This approach ensures that the energy is evenly distributed among the pulses, as thoroughly detailed in [33], forming a flat burst in the GHz-burst regime. These two points represent two extremes of our working range: 50 ppb corresponds to a short burst with moderate intensity per pulse, while 800 ppb corresponds to a long burst with even much lower intensity per pulse, where the cumulative effects are more pronounced. Investigating these extremes allows us to explore how burst length and pulse intensity influence the heat accumulation dynamics and material response during laser ablation.

### 2.2. Machining Workstation

As shown in Figure 1, a beam expander was used to set the diameter of the beam to 5.4 mm at the entrance of the scanner module. The latter included a galvo scanner (LS-Scan XY 20, LASEA, Liège, Belgium) with an f-theta lens of 100 mm focal length, providing a wide field of 60 × 60 mm^2^ and a beam focus of 21.5 µm. For the polishing step, an offset of 600 µm was applied with respect to the focus, leading to a beam diameter of 45.3 µm on the sample. This setup was placed on a motorized Z-axis stage (Alio Industries, Arvada, CO, USA, AI-LM-10000-I-PLT-LP) for precise positioning of the Gaussian beam with respect to the sample surface. Moreover, the station was equipped with a top-view imaging system featuring a long-distance microscope (Edmund Optics, Barrington, NJ, USA, InfiniMax KX with MX5 objective) coupled with a Basler CMOS camera (Basler Inc., Exton, PA, USA, Basler acA1440-220uc, resolution 1440 × 1080, pixel size 3.45 µm × 3.45 µm, 227 fps, global shutter) in order to precisely set the focus position of the beam.

Additionally, the sample was placed inside an argon-fueled overflowing gas cover on an XY-monolithic motorized stage (Alio Industries, AI-LM-20000-XY-I-LP) in order to position it with respect to the scanner and vision system as well as to control the translation for different series of experiments. The entire machining station was mounted on a stiff granite base and gantry to ensure experimental stability and repeatability. The translation stages, laser gate, laser parameters, image recording system, and power modulator unit (composed of a half-wave plate and a polarizing beam splitter) were automated and managed through DMCpro software, v1.0.8 (Direct Machining Control, Vilnius, Lithuania), providing us with complete control and repeatability over the machining station and its parameters.

### 2.3. Methods and Definition

In this section, an explanation of the different methods and applied parameters is provided. The schematic drawing shown in Figure 2 illustrates all the parameters used to compute the accumulative dose. Here, S-2-S is the distance between two consecutive pulses along a line in the x-plane, while L-2-L is the distance between two consecutive lines along the y-plane. Using those two distances, it is possible to retrieve the number of points Nx and Ny located along the X-axis and Y-axis, respectively. Moreover, the burst energy can be computed using the average power and the repetition rate. These parameters allow us to calculate the accumulative dose, which is an essential metric used for comparison in the following sections. The accumulative dose can be calculated as detailed in Equation (Equation 1):(1)$$Dose\left[J\cdot {\mathrm{mm}}^{-2}\right]=E_{burst}*N_{x}*N_{y}$$
with $N_{x}=\frac{1}{L-2-L}$, $N_{y}=\frac{1}{S-2-S}$, and $E_{burst}=\frac{P_{mean}}{freq}$.

### 2.4. Experimental Setup Used for Roughness Measurement

Stainless steel 316L with a thickness of 1 mm was selected for this study. Surface quality was evaluated using the average roughness parameter Ra, which is defined as the arithmetic mean of the absolute deviations of the surface profile from the mean line over a specified distance. While it is one of the most commonly used parameters for quantifying surface finish, particularly in mechanical and manufacturing contexts, Ra only captures vertical surface deviations and does not provide information about the spacing or shape of surface features.

Surface roughness measurements were performed using a confocal microscope (ZEISS, Smartproof 5) equipped with a 20× objective (numerical aperture 0.7), providing a field of view of 0.5 × 0.5 mm^2^. According to the ISO 25178-2 standard, on a real surface texture this field size is insufficient to accurately measure Ra values in the range of 0.1 to 2 µm. To meet the requirements of this standard, each measurement was reconstructed by stitching a 2 × 2 grid of images, resulting in a total analyzed area of 1 mm^2^. The measurement uncertainties are estimated to be ±5 µm for depth measurement with the 10× objective (numerical aperture 0.4) and ±0.1 µm for surface roughness measurements with the 20× objective (numerical aperture 0.7). Note that the errors bars are not visible in the graphs in Section 3, as they are too small.

We calculated Ra values based on eight linear profiles, each at least 800 µm long, taken across the sample and oriented perpendicularly to the hatching direction. Additionally, the Sa parameter, representing the arithmetic mean height over an area, was computed from the entire stitched 1 mm^2^ surface, providing a more comprehensive areal texture characterization.

### 2.5. Gas Cover

As recommended in [21], we applied a cover gas during laser polishing. The gas cover system was fabricated with a circular shape and an open top. The gas was delivered onto the target through eight pipes and a low-density aluminum floor in order to homogenize the gas flow. The flow was adjusted directly with a control valve present in the experimental setup.

### 2.6. Base Surface

The image of the pristine stainless steel 316L sample shown in Figure 3 was acquired using a Scanning Electron Microscope (SEM) at a magnification of 5000× and with a field of view of 55.6 µm. The surface exhibits high smoothness and presents minimal defects. Surface roughness measurements obtained using the confocal microscope yielded a Ra value of 0.06 µm, which is typical for a mirror-like surface. Unless otherwise stated, this surface serves as the baseline for all subsequent experiments.

## 3. Results and Discussion

In this section, two different studies are presented: first, the single-step polishing process on the pristine stainless steel sample, and second, a double-step process based on a first laser milling step followed by a finishing/polishing step on the milled surface. We first present the double-step process (see Section 3.1). Then, we investigate the influence of key processing parameters such as spot-to-spot (S-2-S) and line-to-line (L-2-L) overlap, burst repetition rate, and number of passes on the resulting surface roughness. These effects are analyzed starting from a mirror-like surface, as detailed in Section 3.2, and compared with the results obtained on the samples that were subject to prior milling.

### 3.1. GHz-Burst for Efficient and Smooth Laser Ablation

In this section, we present a proof-of-concept for an all-in-one approach combining ablation and polishing using GHz-bursts. First, we present some results dealing with ablation depth as a function of cumulative dose for short (50 ppb) or long (800 ppb) bursts. The corresponding surface roughness (Ra) results are displayed in Section 3.1.1. The purpose at this stage is to remove matter, not to polish. In a second step, the resulting surface is polished using long GHz-bursts. The influence of the number of passes is analyzed for 100 µm-deep cavities in Section 3.1.2.

#### 3.1.1. Ablation Comparison: Short vs. Long GHz-Bursts

In this section, the ablation depth of short and long GHz-bursts on a pristine 316L stainless steel surface (shown in Figure 3) is presented as a function of the cumulative dose. The change in cumulative dose is linked to a change in the energy contained in a burst ranging from 9 µJ to 94 µJ. The burst length chosen for this study was 50 ppb for the short burst and 800 ppb for the long burst. Each ablation trial was performed by engraving a 2 × 2 mm square, with 165 passes and a hatch of 10 µm rotating by 60° each 55 passes. The burst repetition rate was kept constant at 400 kHz and the marking speed was set to 1 m·s^−1^, providing an S-2-S distance of 2.5 µm. The sample was placed at the focal plane thanks to the off-axis camera; thus, a spot size of 21.5 µm was used and no re-focalization was performed during the process. The surface roughness was determined using the method described in Section 2.4, and the depth of engraving was also determined with the confocal microscope. The resulting depth and roughness measurements are displayed in Figure 4a.

Figure 4a presents the relationship of the engraved depth and surface roughness (Ra) as a function of the cumulative dose, as defined in Equation (Equation 1). The change in cumulative dose is induced through a change in burst energies ranging from 9 µJ to 94 µJ. These energy levels are significantly lower than those studied in [34]. The downward-pointing triangles represent the engraved depth (left black-labeled axis) for burst durations of 50 ppb and 800 ppb, while the crosses correspond to the surface roughness (Ra), which is represented along the right red-labeled axis.

Starting with the depth, we observe that as the dose (and consequently the fluence) increases, the depth of engraving also increases for both curves. However, a clear distinction can be made between the 50 ppb and 800 ppb curves. Specifically, the engraving depth for the 50 ppb curve increases at a rate of 4.75 µm/J·mm^−2^, while the 800 ppb curve shows a much lower rate of 0.85 µm/J·mm^−2^, expressing a completely different interaction with the surface. Indeed, the resulting engraved squares from the 50 ppb and 800 ppb setups exhibit differences in surface morphology. In the case of the 800 ppb configuration, the surface appears melted and “smooth”, whereas in the 50 ppb setup noticeable roughness from the ablation process is present, as shown in Figure 4b and close-up view in Figure 4c.

Turning to the surface roughness, we notice a trend of the surface roughness increasing with the dose, as reported in [32]. In this study, we observed that as the depth of engraving increases (here related solely to the fluence), the surface roughness deteriorates. For the 50 ppb configuration, the surface roughness reaches up to 2.6 µm. With the 800 ppb setup, the surface roughness degrades only slightly up to a dose of 70 J·mm^2^ and the maximum surface roughness is only 0.9 µm, although at the expense of a reduced engraving depth.

For the subsequent section, we selected the laser parameters in order to achieve a cavity depth of 100 µm. This set of parameters corresponds to the encircled point in Figure 4. This is used as the reference surface for the following polishing process with long GHz-bursts (see Section 3.1.2), as illustrated by the SEM images in Figure 4b (with magnification of 50×) and Figure 4c (with magnification of 5000×). In these images, the final pass hatching angle can be observed; zooming in reveals the individual tracks with an observable 50% overlap. Additionally, the individual recast marks generated by each molten pool are visible, contributing to the surface roughness. For minimal roughness, the recast marks would ideally not be visible, producing a single smooth track.

#### 3.1.2. Second Step: Polishing

This section investigates the polishing process applied to previously engraved surfaces (identified in Figure 4a by a purple circle). Using parameters that resulted in an ablation depth of 100 µm as the starting surface, we evaluated how the number of passes of subsequent polishing affected the final surface roughness.

All experiments were conducted at a fixed burst repetition rate of 400 kHz, using GHz-burst mode with 800 ppb and a total burst energy of 46.8 µJ. The laser spot size was 45.3 µm, with a spot-to-spot (S-2-S) distance of 5 µm and a line-to-line (L-2-L) spacing of 1.2 µm.

The results are presented in Figure 5a, which shows the surface roughness as a function of the number of polishing passes. It is important to note that both the engraving and polishing steps were performed using the same burst repetition rate and energy. However, the marking parameters, specifically the burst duration and spot size, were adjusted to achieve the polishing effect (the focus was set 600 µm below the target surface).

Examining Figure 5a, the red dashed line represents the surface roughness after the milling step and before polishing (approximately 1.75 µm). The x-axis indicates the number of polishing passes N, ranging from 1 to 9, used for polishing after the engraving process (described in Section 3.1.1), while the y-axis shows the surface roughness (Ra in blue, Sa in green). The Ra values were measured using the method described in Section 2.4, while the Sa values were extracted directly from the stitched 3D images. The Ra and Sa curves both follow similar trends and show comparable magnitudes, although Sa tends to be slightly higher overall. Figure 5b displays SEM top-view images (5000× magnification) of the surface after each number of passes.

At N = 1, the surface exhibits early signs of melting but largely retains the original morphology in Figure 4c. By N = 2, a significantly smoother surface is achieved, with roughness reduced to just above 0.5 µm. Interestingly, at N = 3 and N = 4, the roughness increases to 1.15 and 1.25 µm, respectively, although this variation is not clearly reflected in the corresponding SEM images. This anomaly is attributed to the formation of thermal surface waviness with a periodicity of a few hundred microns, which subtly increases the measured roughness without drastically altering the visual appearance of the surface.

From N = 5 onward, a clear plateau can be observed; the surface roughness stabilizes just below 0.5 µm and does not significantly improve with additional passes. The SEM images for passes N = 5 to N = 9 reveal the emergence of small white micro-spheres and larger dark spots. These features appear embedded within the surface but do not lead to increased surface roughness. Their exact origin remains undetermined, and further investigations are required to explain their content and appearance.

To summarize this section, GHz-burst laser ablation combined with GHz-burst laser polishing effectively reduces the surface roughness of the engraved samples. The time efficiency of this approach is noteworthy, as only the burst length and spot size were modified between the ablation step and polishing step; key parameters such as intra-burst frequency, burst energy, and burst frequency remained unchanged.

### 3.2. Ghz-Burst for Direct Polishing

This section presents a comprehensive study on the influence of key parameters for laser polishing without any preliminary laser milling. As such, these experiments were carried out on the mirror-polished AISI 316L stainless steel samples, as shown in Figure 3.

Section 3.2.1 examines the effect of spot-to-spot (S-2-S) overlap, which is modulated through variations in burst repetition rate and scanning speed. The influence of the number of passes is analyzed in Section 3.2.2. Finally, Section 3.2.3 explores the scalability of the process.

It is important to note that all experiments were conducted using two distinct line-to-line (L-2-L) distances, providing additional insight into the role of the laser dose as defined in Equation (Equation 1).

#### 3.2.1. Influence of the Spot-to-Spot (S-2-S) Overlap

This section investigates the influence of the spot-to-spot (S-2-S) overlap on surface roughness by systematically varying the burst repetition rate and scanning speed. Experiments were conducted using two distinct line-to-line (L-2-L) overlaps of 1.2 µm and 12 µm. The burst length and burst energy were kept constant at 800 ppb and 2.3 µJ, respectively, with a spot size of 45.3 µm on the sample. The tested repetition rates were 50, 100, 200, 400, and 800 kHz in combination with scanning speeds ranging from 100 to 1400 mm·s^−1^, resulting in S-2-S overlaps ranging from 73% to 99.7%.

The results are presented in Figure 6a for the L-2-L overlap of 12 µm and Figure 6b for the L-2-L overlap of 1.2 µm.

Figure 6a presents the results for an L-2-L overlap of 12 µm. The blue and yellow curves, respectively corresponding to 50 kHz and 100 kHz, show minimal changes in surface roughness across the range of S-2-S overlaps. These two burst repetition rates induce only a slight “cleaning” effect, without any significant surface degradation or improvement.

At 200 kHz (green curve), the initial two data points show relatively low surface roughness. However, a noticeable rise in roughness occurs as the S-2-S overlap increases beyond 87%, eventually plateauing around an Ra of 2 µm. In contrast, the results at 400 kHz and 800 kHz (red and purple curves) reveal a strong negative impact on the surface quality even at the first test points; at overlaps of 93% and 96%, the surface roughness already reaches 5 µm and 2.8 µm, respectively. As the overlap increases further, the surface roughness escalates dramatically, reaching values as high as 11 µm and 13 µm for S-2-S overlaps exceeding 96% and 98%. This trend indicates a transition to a more thermally-driven interaction regime, leading to pronounced surface degradation.

Turning to Figure 6b, which corresponds to an L-2-L overlap of 1.2 µm, a different behavior is observed. As in the previous case, the 50 kHz and 100 kHz curves show negligible impact, with formation of a thin uniform melt layer that is insufficient for effective polishing. At 200 kHz, the surface initially exhibits a sharp increase in roughness at low overlaps. This is followed by a decrease to an Ra of approximately 2.5 µm at an S-2-S overlap of 84%, before rising again to about 4 µm at 98% overlap. This fluctuation is attributed to formation of a recast layer at low overlap values and to thermal instabilities at higher ones. The smaller L-2-L spacing of 1.2 µm exacerbates the roughness by pushing the molten material to the track edges, leading to localized accumulation and increased surface irregularity.

Finally, for 400 kHz and 800 kHz, both curves show similar trends. The surface roughness starts high, at around 3 µm and 2.7 µm, respectively, and remains roughly constant as the S-2-S overlap increases. Unlike in Figure 6a, where the roughness increases sharply at these burst repetition rates, the wider L-2-L spacing results in less pronounced thermal accumulation. However, the higher overlap also exhibits some thermal instability, yielding persistently poor surface quality.

#### 3.2.2. Influence of the Number of Passes

This section explores the effect of the number of passes on the final surface roughness. Experiments were carried out using two different line-to-line (L-2-L) distances: 1.2 µm and 12 µm. The burst length was fixed at 800 ppb, and two burst energies of 2.3 µJ and 46.8 µJ were tested. Burst repetition rates of 50, 100, 200, 400, and 800 kHz were evaluated in combination with scanning speeds ranging from 250 to 4000 mm·s^−1^, resulting in a constant spot-to-spot (S-2-S) overlap of 91.6% (i.e., a spacing of 5 µm).

The surface roughness results are shown in Figure 7a for the specific case of 400 kHz burst repetition rate, L-2-L = 1.2 µm, and a burst energy of 46.8 µJ. Figure 7b depicts the results for a burst energy of 2.3 µJ at different burst repetition rates for L-2-L = 12 µm, and Figure 7c depicts the same for L-2-L = 1.2 µm.

Figure 7a illustrates the effect of the number of passes on the surface roughness using the parameters defined in Section 3.1.2. The red dashed line represents the initial surface roughness. In contrast to the results from Section 3.1.2, where the surface roughness was reduced with increasing passes, it can be seen that Ra worsens progressively. This indicates a different polishing behavior under identical parameters, depending on the initial surface condition. Additionally, the Ra and $S_{a}$ values match closely throughout, validating the consistency of our surface roughness measurements.

Figure 7b shows the evolution of surface roughness for an L-2-L distance of 12 µm. Across all tested burst repetition rates, a consistent three-phase trend can be observed. Initially, up to N = 3, the roughness values exhibit high variability, ranging from 0.22 µm to 3.7 µm. Between N = 3 and N = 6, a stable plateau forms, with Ra values between 0.22 µm and 0.55 µm, indicating temporary stabilization of the surface quality. Beyond N = 6, a new regime emerges marked by a gradual increase in roughness; for example, at 400 kHz and N = 8, the roughness reaches a maximum of 6.4 µm.

In contrast, Figure 7c presents the results for a smaller L-2-L spacing of 1.2 µm (i.e., a higher dose). Here, the surface roughness increases immediately from N = 1, with Ra values ranging from 1.5 µm to 5.7 µm. For burst repetition rates of 50, 100, 200, and 400 kHz, the roughness decreases until around N = 4, reaching a minimum of approximately 0.5 µm for 100 kHz. However, different behavior is present at 800 kHz; the roughness rises sharply, peaking at 27 µm by N = 4. After this point, the trends diverge. While the 200 kHz and 400 kHz curves continue to rise steadily, reaching 19 µm and 16 µm, respectively, by N = 8 the 800 kHz curve unexpectedly decreases to 2.78 µm at the same pass number.

From the data presented in this section, it can be inferred that the effectiveness of the polishing process depends strongly on the initial surface state (absorption and reflectivity). Although identical parameters were used, starting from a mirror-like surface (as in Figure 7c) resulted in an increase in roughness, whereas the same settings applied to a previously ablated surface (Section 3.1.2) led to the formation of a molten layer and reduction in Ra below 0.5 µm with multiple passes.

Furthermore, comparison between Figure 7a,b suggests that lower initial surface roughness benefits more from a reduced dose (achieved here through larger L-2-L spacing), which appears to promote the formation of a smoother molten layer and better surface finish.

#### 3.2.3. Scalability

This section investigates the scalability of the polishing process with respect to the final surface roughness. Experiments were conducted using two different line-to-line (L-2-L) distances of 1.2 µm and 12 µm. The burst length was fixed at 800 ppb, with burst energies ranging from 9.8 to 93.75 µJ.

Five different burst repetition rates of 50, 100, 200, 400, and 800 kHz were assessed in combination with scanning speeds between 100 and 1400 mm·s^−1^ while maintaining a constant spot-to-spot (S-2-S) overlap of 91.6 % (equivalent to a 5 µm spacing).

The results are presented in Figure 8a for L-2-L = 1.2 µm and in Figure 8b for L-2-L = 12 µm. Additionally, the SEM images corresponding to Figure 8a are provided in Figure 9.

Figure 8a illustrates the evolution of surface roughness for an L-2-L distance of 1.2 µm as a function of the deposited dose, which was adjusted by varying the burst energy. Across all tested burst repetition rates, a clear trend is evident: from 0.5 J·mm^−2^ to approximately 3.3 J·mm^−2^, the surface roughness values increase gradually, reflecting minor surface modifications; however, in the range of 3.8 J·mm^−2^ to 4.3 J·mm^−2^, a higher variability is observed across all burst repetition rates, indicating a critical energy threshold beyond which the process becomes less stable. Above 4.3 J·mm^−2^, the increasing dose leads to significant thermal instabilities, causing pronounced roughness due to excessive localized heating. These observations confirm that the scalability is generally maintained for this L-2-L distance, as similar outputs are obtained across varying burst repetition rates when the S-2-S overlap remains constant, suggesting that process acceleration through higher repetition rates is feasible.For L-2-L = 12 µm, as shown in Figure 8b, the surface roughness follows a similar pattern. The roughness gradually increases for doses from 0.5 J·mm^−2^ to 5.4 J·mm^−2^, demonstrating consistent scalability, as the outputs remain comparable across burst repetition rates under the same S-2-S overlap condition.

Finally, Figure 9 offers a more detailed view of the surface morphology across varying energy levels. In the blue-highlighted region, representing 0.5 J·mm^2^ for all repetition rates, minimal surface alteration is observed; however, the red-highlighted region, corresponding to higher energy levels (3.3 J·mm^2^ to 5.4 J·mm^2^), reveals significant thermal instability, including micro-sphere formation and track wobbling, indicative of excessive energy transfer. Interestingly, the intermediate region exhibits the smoothest track formation, particularly for 400 kHz and 800 kHz, although edge effects still induce artificial roughness. This limitation could be mitigated with advanced beam shaping techniques such as annular beam profiles in order to reduce the central peak intensity and improve the overall uniformity.

## 4. Conclusions

In conclusion, this study has thoroughly demonstrated the feasibility and scalability of GHz-burst laser polishing for enhancing the surface quality of metallic materials. The initial proof-of-concept confirmed the fundamental viability of the GHz-burst regime, highlighting the ability of this approach to effectively control the surface morphology and demonstrating its potential to reduce roughness without excessive thermal damage when parameters are precisely controlled. Our studies on the number of passes underline the importance of controlling the cumulative energy input. Results show that while multiple passes can significantly enhance surface finish, the required number of passes is highly dependent on the initial surface roughness, showing a tenfold increase in the burst energy required to smooth a much rougher surface (Ra of 1.75 µm). However, excessive repetition of passes can introduce thermal stresses, potentially compromising the material’s integrity. This insight is critical for applications with stringent surface specifications, where precision is paramount. Finally, our scalability investigation demonstrated the adaptability of GHz-burst laser polishing across varying repetition rates and marking speeds, achieving a 16× proportional speed increase with 800 kHz at 1.4 m.s^−1^ without sacrificing surface quality. These results pave the way for integrating this technology into high-throughput manufacturing systems, addressing critical industry demands.Collectively, these findings establish GHz-burst laser polishing as a powerful and adaptable tool for next-generation manufacturing, with potential applications spanning biomedical implants, aerospace components, and precision optics. Moving forward, efforts should focus on refining energy distribution strategies, integrating real-time feedback systems, and exploring advanced beam shaping in order to further enhance performance, minimize edge effects, and unlock the full potential of this innovative approach. 

## Figures and Tables

**Figure 1 nanomaterials-15-01343-f001:**
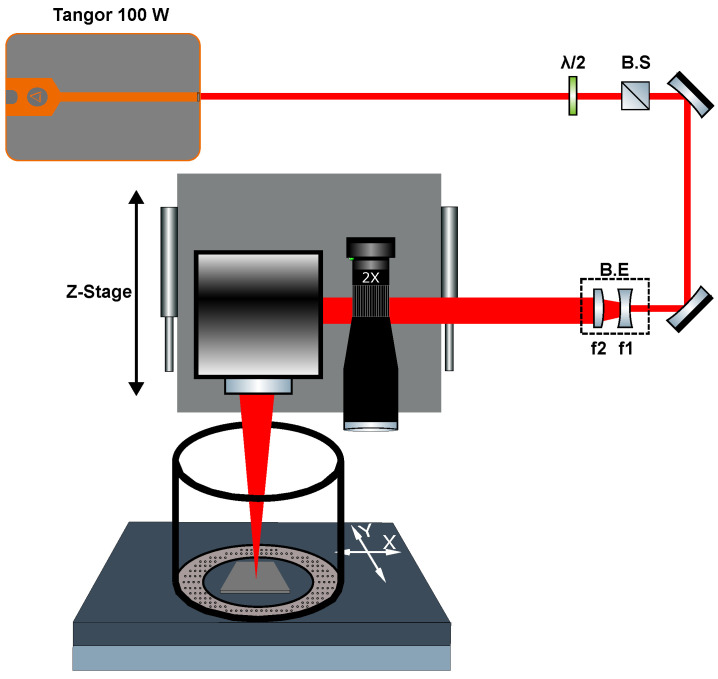
Experimental setup used for metal polishing using GHz-burst irradiation; B.S. = Beam Splitter, B.E. = Beam Expander with, f1 = 50 mm and f2 = 100 mm.

**Figure 2 nanomaterials-15-01343-f002:**
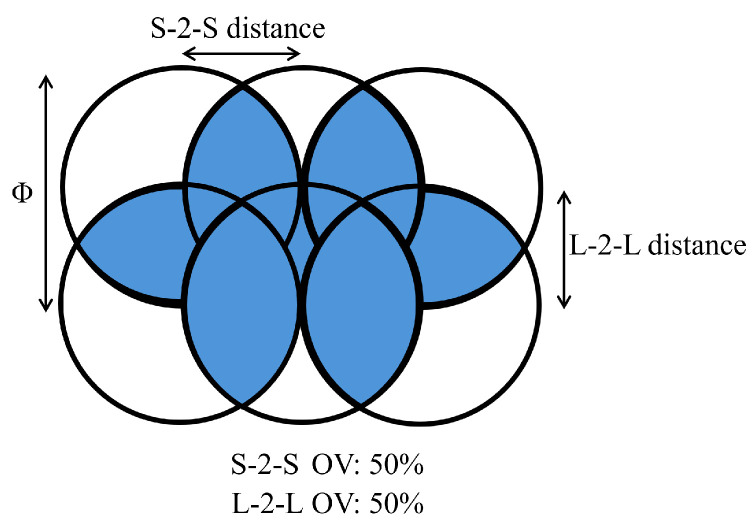
Schematic of laser tracks for specific S-2-S and L-2-L distance.

**Figure 3 nanomaterials-15-01343-f003:**
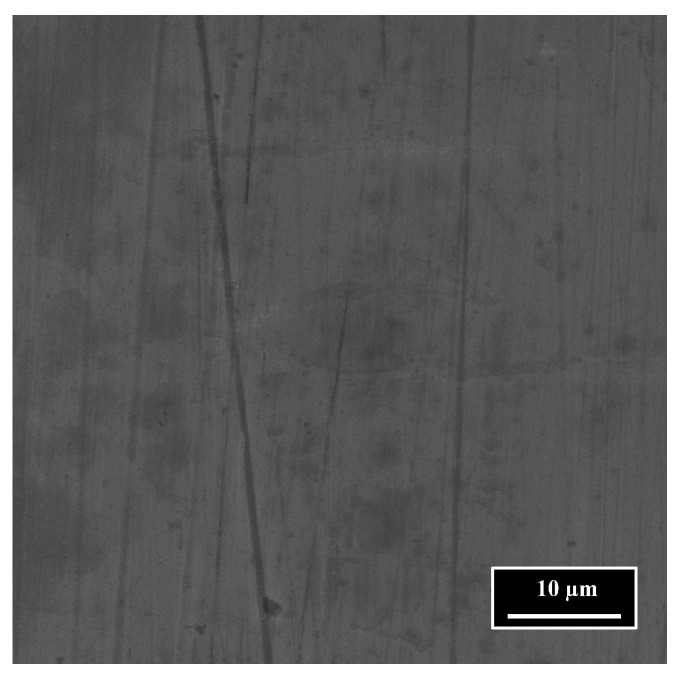
SEM top view of pristine 316L surface with magnification of ×5000.

**Figure 4 nanomaterials-15-01343-f004:**
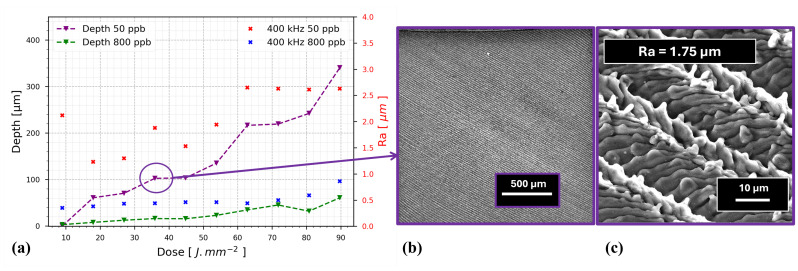
(**a**) Left axis: Evolution of the engraved depth obtained in 316L steel with short (purple triangles) and long (green triangles) GHz-bursts as a function of the cumulative dose. Right axis: Evolution of the surface roughness, Ra ISO (25178-2) for short (red squares) and long (blue squares) GHz-bursts as a function of the dose. Errors are estimated to 5 µm for depth measurements. (**b**) SEM top view of 316L steel surface obtained after laser irradiation with 50 ppb for a dose of 36 J·mm^2^ at magnification of ×136. (**c**) SEM top view of 316L steel surface obtained after laser irradiation with 50 ppb for a dose of 36 J·mm^2^ at a magnification of ×5000.

**Figure 5 nanomaterials-15-01343-f005:**
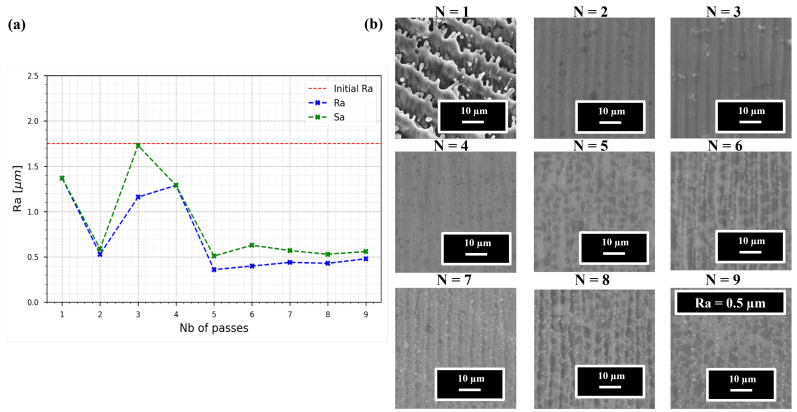
(**a**) Evolution of the surface roughness Ra ISO (25178-2) as a function of the number of passes on 316L steel for 800 ppb GHz-bursts at 400 kHz, 46.8 µJ per burst, S-2-S = 5 µm, L-2-L = 1.2 µm. (**b**) SEM top view of the 316L surface obtained after laser irradiation with 800 ppb at magnification of ×5000.

**Figure 6 nanomaterials-15-01343-f006:**
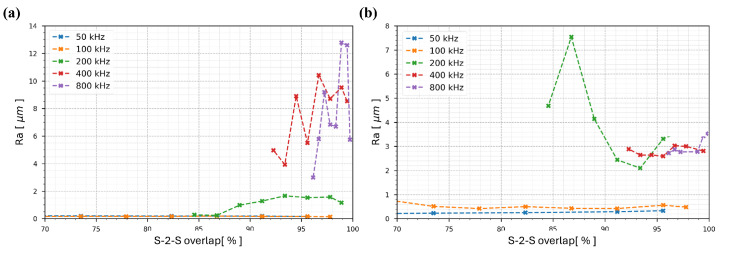
Surface roughness Ra variation as a function of the S-2-S overlap for 316L steel samples under GHz-burst irradiation with 800 ppb and 2.3 µJ per burst, with (**a**) L-2-L = 12 µm and (**b**) L-2-L = 1.2 µm.

**Figure 7 nanomaterials-15-01343-f007:**
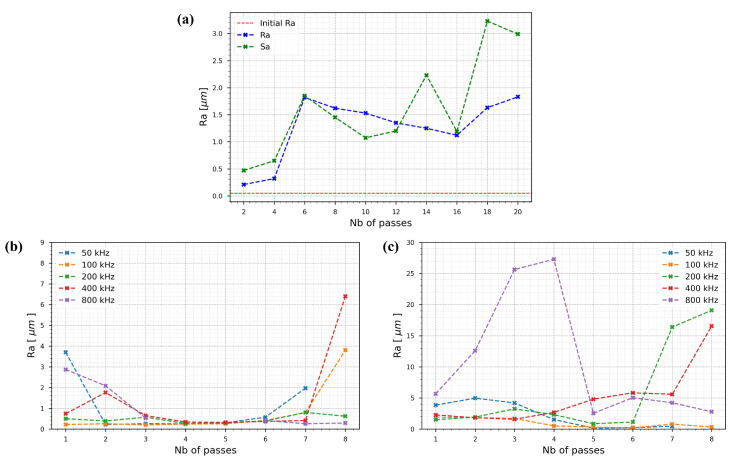
Surface roughness Ra evolution as a function of the number of passes for 316L steel samples under GHz-burst irradiation with 800 ppb and S-2-S = 5 µm for: (**a**) L-2-L = 1.2 µm, 400 kHz, 46.8 µJ per burst (the red dashed line indicates the initial surface roughness) (**b**) L-2-L = 12 µm, 2.3 µJ per burst; and (**c**) L-2-L = 1.2 µm, 2.3 µJ per burst.

**Figure 8 nanomaterials-15-01343-f008:**
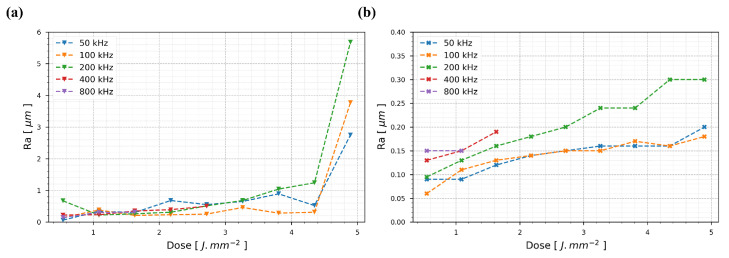
Evolution of surface roughness Ra as a function of dose in 316L stainless steel, with S-2-S = 5 µm and 800 ppb GHz-burst: (**a**) for L-2-L = 1.2 µm and (**b**) for L-2-L = 12 µm.

**Figure 9 nanomaterials-15-01343-f009:**
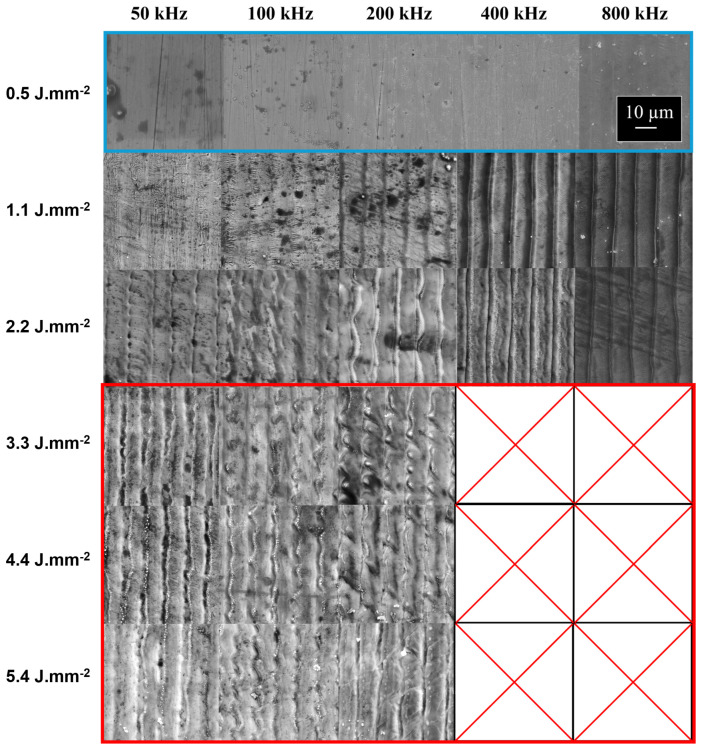
SEM top view of 316L steel sample at ISO dose with S-2-S = 16 µm, L-2-L = 12 µm, 800 ppb GHz-burst, 2.3 µJ per burst for multiple burst repetition rates, and magnification of ×5000.

## Data Availability

The original contributions presented in this study are included in the article. Further inquiries can be directed to the corresponding authors.

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
