# Peer review of "Long GHz-Burst Laser Surface Polishing of AlSl 316L Stainless Steel Parts Manufactured by Short GHz-Burst Laser Ablation"

_nanomaterials, 2025, doi:10.3390/nano15171343_

Round 1

Reviewer 1 Report

Comments and Suggestions for Authors

This manuscript investigates the application of GHz-burst laser polishing technology for surface treatment of AISI 316L stainless steel. It compares the effects of single-step and double-step processes, systematically analyzes the influence of key parameters on surface roughness. The content is substantial and comprehensible, but the following minor revisions need to be made before publication:

1. The paper selects 50 ppb (short pulse) and 800 ppb (long pulse) as the two extremes of burst length but does not explain why these two values were chosen. It is suggested to supplement parameter screening with preliminary experimental data or theoretical basis to clarify their representativeness.

2. Figure 4 only shows the SEM image after laser irradiation with 50 ppb, while the laser irradiation with 800 ppb is only described as “smooth” without providing an SEM image. The corresponding SEM image should be provided for comparison.

3. In the second-step polishing, why choose a surface with a roughness of 1.75 um as the engraving surface instead of a surface with a smaller roughness?

4. Under identical parameters, different initial surface conditions exhibit different polishing behavior. This should be explained.

5. The surface roughness obtained by double-step polishing is comparable to that obtained by the first step using 800 ppb GHz pulse ablation, approximately 0.5 μm. What are the advantages of double-step polishing?

Reviewer 2 Report

Comments and Suggestions for Authors

(1) There is no indication of the number of repetitions conducted.

(2) Error bars should be added in Ra results.

(3) In Figure 7c, some Ra values were not visible, please clarify.

(4) In Figure 9, the magnifications of SEM test should be supplied.

(5) How the L-2-L influence the Ra values, the authors are encouraged to give a more scientific explanation.

(6) The conclusions should be concise and specified.

Comments on the Quality of English Language

The English could be improved to more clearly express the research.

Reviewer 3 Report

Comments and Suggestions for Authors
  1. There are some grammar issues in the entire text, please revise them in a timely manner.
  2. The paper successfully demonstrates the application of GHz-burst laser in polishing and milling 316L stainless steel surfaces, particularly highlighting the feasibility of a "two-step process" (milling + polishing), which shows certain innovativeness. However, the Introduction does not sufficiently emphasize how this work differs from existing GHz-burst laser studies (e.g., [25]–[31]). It is recommended to more clearly articulate the unique contributions of this study.
  3. Figures 4a, 5a, 6, 7, and 8 provide rich data, but error bars are missing, and some data points (e.g., Ra values) show significant fluctuation, lacking tests of statistical significance (e.g., standard deviation or confidence intervals).
  4. The use of stitched confocal images to comply with ISO 25178-2 is reasonable. However, the number of measurement repetitions (e.g., n≥3) or consistency across different regions is not specified.
  5. The conclusion section is too long, please compress it.
